# The Influence of Silver-Containing Bionanomaterials Based on Humic Ligands on Biofilm Formation in Opportunistic Pathogens

**DOI:** 10.3390/nano14171453

**Published:** 2024-09-06

**Authors:** Maria V. Zykova, Maria R. Karpova, Yu Zhang, Marianna V. Chubik, Daria M. Shunkova, Lyudmila A. Azarkina, Dmitrii A. Mihalyov, Andrey I. Konstantinov, Evgenii V. Plotnikov, Alexey N. Pestryakov, Irina V. Perminova, Mikhail V. Belousov

**Affiliations:** 1Pharmaceutical Faculty, Siberian State Medical University, Tomsk 634050, Russia; karpova.mr@ssmu.ru (M.R.K.); chubik.mv@ssmu.ru (M.V.C.); shunkova.dm@ssmu.ru (D.M.S.); ludmila_logvinova@mail.ru (L.A.A.); diman021999@gmail.com (D.A.M.); mvb63@mail.ru (M.V.B.); 2Department of Chemistry, Lomonosov Moscow State University, Leninskiye Gory 1-3, Moscow 119991, Russia; zhangyu13051837552@gmail.com (Y.Z.); noogen@inbox.ru (A.I.K.); iperminova@gmail.com (I.V.P.); 3Research School of Chemistry and Applied Biomedical Sciences, Tomsk Polytechnic University, Tomsk 634050, Russia; plotnikovev@tpu.ru (E.V.P.); pestryakov@tpu.ru (A.N.P.)

**Keywords:** humic substances, phenol–humic derivatives, silver nanoparticles, opportunistic pathogens, antibacterial activity, biofilm formation

## Abstract

The uncontrolled use of antibiotics has led to a global problem of antimicrobial resistance. One of the main mechanisms of bacterial resistance is the formation of biofilms. In order to prevent the growth of antimicrobial resistance, it is crucial to develop new antibacterial agents that are capable of inhibiting the formation of biofilms. This makes this area of research highly relevant today. Promising candidates for these antibacterial agents are new bionanomaterials made from natural humic substances and silver nanoparticles. These substances have the potential to not only directly kill microorganisms but also penetrate biofilms and inhibit their formation. The goal of this study is to synthesize active pharmaceutical substances in the form of bionanomaterials, using ultradispersed silver nanoparticles in a matrix of coal humic substances, perform their characterization (NMR spectroscopy, TEM, and ICP-AES methods), and research their influence on biofilm formation in the most dangerous opportunistic pathogens (*E. coli*, *Methicillin-resistant St. Aureus*, *K. pneumoniae*, *P. aeruginosa*, *St. aureus*, *A. baumannii*, and *K. Pneumonia*). The results showed that all of the studied bionanomaterials had antibacterial activity against all of the opportunistic pathogens. Furthermore, they were found to have a suppressive effect on both pre-existing biofilms of these bacteria and their formation.

## 1. Introduction

The discovery and development of technology for industrial production of antibiotics in the 1940s revolutionized the treatment of bacterial infections, leading to a significant decrease in deaths and an increase in human lifespan by an average 10 years [1,2,3,4]. Antibiotics’ efficacy against human pathogens, in addition to animal and plant pathogens, as well as their ability to stimulate key biochemical processes of adaptation to unfavorable conditions, contributed to their increasing use not only for the treatment of infectious diseases but also for their prevention, the stimulation of growth and formation of phytomass, and increasing productivity in veterinary medicine and agriculture [5,6,7]. The valuable biological effects of antibiotics, with their relatively simple and cost-effective ways of production, could, in the opinion of many scientists, policymakers, and businessmen, make them one of the ways to solve the global problem of food scarcity [8]. But in the end, the uncontrolled use of antibiotics has created a problem no less dangerous than global food scarcity, antimicrobial resistance (AMR). Despite its recent emergence, the issue of AMR has the status of a worldwide threat due, on the one hand, to the inability to stop growth in the number of patients whose isolates contain AMR strains, but, on the other hand, to the emergence of multidrug-resistant organisms, the probability of infection with which is not limited only to stays in hospitals and intensive care units [1,9]. The WHO Is concerned that without appropriate measures to curb AMR, a heavy financial burden of approximately USD 100 trillion might be loaded on the global GDP, and over 10 million patients might be threatened by premature death [9,10]. To prevent a return to the “pre-antibiotic era”, the WHO and the agencies under its control have adopted several strategic measures, including strengthening the monitoring the use of antibiotics for the treatment of infectious diseases in medicine and agriculture, tightening the legislative regulation of antibiotic use, as well as the search for their rational prescription due to improved diagnosis of infections, and wide implementation of measures to prevent infectious diseases (vaccination, water purification measures, maintenance of proper hygiene in medical institutions, agricultural complexes, livestock farms, etc.) [1,11,12,13,14]. However, physicians of different profiles see a vital approach to preventing the growth of AMR in the development of new antibiotics, which makes this direction very relevant today [11]. Infections caused by antibiotic-resistant microorganisms cause about 700,000 patient deaths each year [15]. The most dangerous microorganisms that lead to premature death are *Escherichia coli*, *Staphylococcus aureus*, *Klebsiella pneumoniae*, *Pseudomonas aeruginosa*, *Acinetobacter baumannii,* and other bacteria [16]. All these bacteria, due to their high resistance (listed as “Bacterial Priority Pathogens” by the WHO), pose a global problem for the health and wealth of the entire world population. For this reason, the development of new antimicrobial agents is required to effectively combat them [17].

One of the mechanisms of bacterial resistance is the formation of biofilms. Biofilms are a form of bacterial existence in the form of microbial communities adhered to biotic and abiotic surfaces. In a biofilm, bacteria are embedded in an extracellular matrix (ECM), which allows microorganisms to be resistant to various factors, including antibiotics. Biofilms on the surface of implants, prostheses, medical devices, and equipment (catheters, artificial heart valves, contact lenses, and dental units), when ingested into the body, contribute to the formation of chronic and recurrent infectious processes [18,19]. For this reason, it is very important to search for new antibacterial agents capable of inhibiting the formation of biofilms. One of the priority areas today is the use of silver nanoparticles due to their extensive antibacterial, antimycotic, and antiviral properties; biocompatibility; and effectiveness against microorganisms with multiple drug resistance [20,21,22,23]. Various approaches are used to synthesize metal nanoparticles, but the most environmentally friendly and effective method is the reduction of a metal-containing precursor in an aqueous medium in the presence of a stabilizer of the resulting nanoparticles, including various plant ligands [23,24,25,26,27,28,29] and humic substances [30,31]. Thus, bionanomaterials with silver nanoparticles ultradispersed in humic substance (HS) matrices may be promising agents to overcome antibiotic resistance. Humic substances are defined as dark-colored organic compounds that can be derived from such natural resources as peat, brown coal, sapropel, and others. Chemically, HSs are complex assemblies of molecules [32] that incorporate structures with both polymeric and supramolecular characteristics, lacking a strict constancy in chemical composition. Due to the presence of a large number of functional groups in their composition, HSs exhibit numerous biological effects, mainly immunostimulating and antioxidant effects [33,34,35,36,37,38,39]. Due to the presence of electron-donating groups in the molecule, HSs are able to exhibit affinity for many ions. The combination of biomacroligand and polyelectrolyte properties allows the use of humic substances as stabilizers of nanoparticles and for the synthesis of coordination compounds [40]. Therefore, it is possible to synthesize highly stable nanoparticles of various biogenic metals in their medium, including silver. Silver is of great importance for medicine, as it has antivirulence activity and is unable to cause the development of dysbiosis (a frequently occurring undesirable side effect of antibiotics) [41]. It is known that in soil, biogenic metals are in bound form (in the form of complexes with acid groups of HSs), but at the same time, they are easily available for absorption by microorganisms [42]. Therefore, it can be hypothesized that HSs as ligands for silver nanoparticles will also be highly bioavailable molecules to pass through the ECM biofilms. Stability, nanoscale size, and high reactivity determine the high efficacy of silver nanoparticles against microbes even in low concentrations [43]. The mechanism of antibacterial action of metal nanoparticles is usually described in the literature as cell membrane integrity leakage, disruption of the electron transport chain, reactive oxygen species production, DNA damage, and protein deactivation [23,43,44,45,46,47]. In particular, the mechanism of action of silver nanoparticles is related to their ability due to electrostatic forces to adhere and accumulate in the cell wall of bacteria (silver ions (Ag(I)) have a positive charge, and the cell wall is negatively charged), causing structural changes and its destabilization. Once inside the bacterial cell, Ag(I) interacts with proteins, lipids, and nucleic acids, causing free-radical oxidation reactions. Formed reactive oxygen species and other free radicals cause oxidative stress, disrupt ribosomes, and damage DNA. As a result, the cell dies [23,43,44,45,46,47]. Gram-negative bacteria are more sensitive to silver nanoparticles than Gram-positive bacteria, which is a consequence of the difference in cell wall structure [43]. Due to the thick layer of peptidoglycan, silver nanoparticles are fixed in a certain place on the surface of the bacterial cell for some time. At the adsorbed site, Ag(I) disrupts the transport of ions, carbohydrates, and proteins, which gradually leads to the disruption of cell wall integrity [23,43,44,45,46,47].

Thus, bionanomaterials based on natural HSs and silver nanoparticles are promising candidates for the role of antibacterial agents, potentially able not only to kill microorganisms directly but also to penetrate into biofilms and inhibit their formation.

The possibility of creating such a combination of substances will subsequently help to solve such an important and practically significant social problem as antibiotic resistance.

The goal of this study is the synthesis of active pharmaceutical substances as bionanomaterials based on silver nanoparticles ultradispersed in a matrix of HSs, their characterization, and researching their influence on biofilm formation in opportunistic pathogens.

## 2. Materials and Methods

### 2.1. Synthesis of Humic Acid Derivatives

Sodium salt of coal humic acid (CHP-Na) was used as the initial humic matrix. To obtain derivatives, it used 1,4-dihydroxybenzene (hydroquinone), 2-methyl-1,4-dihydroxybenzene (2-methylhydroquinone), 1,2-dihydroxybenzene (pyrocatechol), 1,4-naphthoquinone, and 2-hydroxy-1,4-naphthoquinone. Before starting the synthesis, sodium humate was freed from ballast (insoluble part) by centrifugation. To do this, a sample of potassium humate (1 g) was dissolved in 35 mL of distilled water and centrifuged for 10 min at 11,000 rpm. 

#### 2.1.1. Synthesis of Phenol-Enriched Derivatives of HSs Using the Fenton Reaction

The synthesis of Fenton derivatives was conducted in accordance with [48]. In brief, the solution of CHP-Na was transferred into a 100 mL beaker, into which 0.25 g of hydroquinone or naphthoquinone was previously added. The pH of the resulting solution was adjusted to 10–11 using a 40% NaOH solution. Then, 2 mL (0.02 mol) of 30% H_2_O_2_ was added, after stirring, and then a solution of iron (II) sulfate (0.3 g (0.001 mol) FeSO_4_ in 10 mL of distilled water) was added dropwise with constant stirring, maintaining the pH between 9 and 10 by adding alkali. The resulting reaction mixture was transferred to a 100 mL flask equipped with a reflux condenser and heated for 4 h at a temperature of 70 °C in a water bath with constant stirring using a thermostatically controlled magnetic stirrer. After completion of the reaction, the reaction mixture was allowed to cool. Then, the flask with the product was placed on a rotary evaporator and dried at a temperature of 50 °C.

#### 2.1.2. Phenol–Formaldehyde Copolycondensation

The synthesis of phenol–formaldehyde derivatives was conducted in accordance with the technique described in [49]. The resulting CHP-Na was transferred into a 100 mL beaker, into which 0.25 g of hydroquinone and 0.1 g of oxalic acid were previously added using the tip of a spatula. Then, a 35% formaldehyde solution was added to the reaction mixture in a ratio of 1 g of solution (12 mmol) per 15 mmol of phenolic hydroxyls. It was assumed that 1 g of CHP-Na contains 5 mmol of phenolic hydroxyls. The mixture was boiled while stirring for an hour, then dried on a rotary evaporator at a temperature of 65 °C. The resulting product was ground and washed out with distilled water by repeated centrifugation and decantation of the wash water. The washed preparation was dried on a rotary evaporator at a temperature of 60 °C. The dried sample was placed into a desiccator over P_2_O_5_ for at least 3 days.

### 2.2. Synthesis and Characterization of Silver Nanoparticles

Sodium salt of coal humic acid (CHP-Na) and phenol-enriched derivatives of sodium humate obtained with the use of the two techniques described above were used as the initial humic matrix (CHP-X-Y), where X is the derivative code (*o-*hydroquinone; *p-*hydroquinone; 2-methyl-1,4-hydroquinone; 1,4-naphthoquinone; 2-hydroxy-1,4-naphthoquinone) and Y the reaction (Fenton; FFK). Synthesis of AgNPs was conducted with a use of conventional and microwave heating as described in [50].

Prior to synthesis, sodium CHP-Na and phenol-enriched sodium humate derivatives were freed from ballast (insoluble part) by centrifugation. To do this, a sample of CHP-Na or phenol-enriched sodium humate derivatives (1 g) was dissolved in 35 mL of distilled water and centrifuged for 10 min at 10,000 rpm. The resulting supernatant was transferred into a 150 mL beaker, and the volume of the resulting CHP-Na solution was adjusted to 70 mL (approximately 35 mL of deionized CHP-Na was added). The pH of the resulting solution was adjusted to 11 using a 0.5 M NaOH solution. Then, a solution of silver nitrate (0.58 g of AgNO_3_ in 13 mL of distilled water) was added dropwise with constant stirring, maintaining the pH in the range of 9.8–10.0 by adding NaOH alkali. The resulting reaction mixture was transferred into a 150 mL flask equipped with a reflux condenser and heated for 4 h at a temperature of 80 °C in a water bath (in dark conditions) with constant stirring using a thermostatically controlled magnetic stirrer.

Microwave synthesis (CHP-AgNPs-MW) was carried out in a modified 800 W household microwave oven using a 15 s pulse sequence with a 10 s pause (on/off 10 s) to prevent boiling and spattering of the solution due to overheating. The total synthesis time was 240 s. After completion of the synthesis, the samples were frozen and dried in a vacuum. All syntheses were carried out in the dark. Finally, the samples were frozen and dried [50]. 

The particles obtained by the methods described above were examined using the transmission electron microscopy (TEM) method, for which a JEOL JEM-2100F microscope (JEOL, Akishima, Japan) was used. Image processing was performed using ImageJ1.54d software. For the synthesized particles, in addition to electronic properties, morphological characteristics (size and shape), form of occurrence, and silver content were established. The total silver concentration was determined by the ICP-AES method using the axial ICP-AES 720-ES spectrometer (Agilent Technologies, Santa Clara, CA, USA).

### 2.3. ^13^C NMR Spectroscopic Study of Hydroxylated Derivatives

The structural-group compositions were determined using ^13^C NMR spectroscopy. ^13^C NMR spectra were recorded in 0.3 M NaOD/D_2_O (99+% isotopic purity, Sigma Aldrich, Burlington, MA, USA). The spectra were recorded on an Avance-400 NMR spectrometer (Bruker, Berlin, Germany) with a carrier frequency for ^13^C nuclei of 100 MHz. The INVGATE pulse sequence was used to exclude the nuclear Overhauser effect. A relaxation delay of 8 s was used for the complete relaxation of quaternary carbon atoms. To calculate the structural-group composition, the spectra were divided into nine intervals corresponding to the main structural components of HSs, and the intervals were integrated. The obtained integrals normalized to the whole spectrum represent the quantitative data of the structural-group composition of the studied HS [51].

### 2.4. Preparation of Microbial Cultures

For the study, standard strains of opportunistic microorganisms were used: *Escherichia coli*, ATCC 25922; *Staphylococcus aureus*, ATCC 25923; MRSA (*Methicillin-resistant Staphylococcus aureus*), ATCC 33592; *Klebsiella pneumonia*, ATCC 700603; *Pseudomonas aeruginosa*, ATCC 9027; and clinical isolates of these types of bacteria, *Escherichia coli*, *Staphylococcus aureus*, *Acinetobacter baumannii*, *Pseudomonas aeruginosa*, and *Klebsiella pneumonia*, isolated in the bacteriological laboratory of Siberian State Medical University clinics from various patient materials (urine, sputum, wounds, the uterine cavity, and blood).

To study the antibacterial effect, 12 samples of the active pharmaceutical substances (HS-AgNPs) were taken, which are bionanomaterials based on silver nanoparticles ultradispersed in a matrix of humic substances.

Pure overnight cultures of the test bacteria were grown on Mueller–Hinton agar (OXOID) in a thermostat at 37 °C.

Bacterial suspensions were prepared in sterile Mueller–Hinton broth (OXOID). The concentration was determined by the optical density of the suspension using a densitometer “DEN-1B” (Biosan). For biofilm formation studies, suspensions were prepared with an optical density of 0.5 McFarland units (1.5 × 10^8^ CFU/mL), and for screening substances for antimicrobial properties, suspensions with an optical density of 2 McFarland units (6.0 × 10^8^ CFU/mL) were used.

### 2.5. Studying the Ability of Microorganisms to Form Biofilms

The biofilm-forming ability of bacteria was determined using the method of Kabanova A.A. “Method for assessing the ability of microorganisms to form biofilms: patent 17673 of the Republic of Belarus”, 2013 [52].

Bacterial suspensions (150 µL) were added to wells of a 96-well flat-bottomed polystyrene cell culture plate (HTI (High Technology Inc.), North Attleboro, MA, USA). As a negative control, 150 µL of pure Mueller–Hinton broth was used. The plates were incubated at 37 °C for 24 h. The contents were then removed from the wells, which were washed four times with distilled water. The wells were then treated with 170 µL of 0.25% alcohol solution of crystal violet (CV) and incubated at room temperature for 5 min. The plates were washed again four times with distilled water and air-dried at room temperature for 10 min. The wells were treated with 200 µL of dimethyl sulfoxide (DMSO) as an extracting agent and incubated at room temperature for 10 min for complete dye extraction. The optical density of the well contents was measured at 620 nm wavelength using a multi-channel spectrophotometer and compared with the optical density of the negative control. Biofilm formation was registered by an increase in the optical density of the experimental wells.

To assess the influence of HS-AgNPs on biofilm formation, HS-AgNPs and Mueller–Hinton broth were added to bacterial suspensions at or below their minimum inhibitory concentration (MIC) to prevent their bactericidal effects. To evaluate the effects of antibacterial preparations on formed biofilms, HS-AgNPs and Mueller–Hinton broth were added 1 h after the first washing of plate wells.

### 2.6. Determination of the Sensitivity of Microorganisms to Nanocomposites

Sensitivity testing of microorganisms to nanocomposites was performed according to MUK 4.2.1890-04 “Determination of microorganism sensitivity to antibacterial agents” in 96-well flat-bottomed polystyrene plates (SPL Life Sciences Co., Ltd.). The wells of the plates received 142 µL of pure Mueller–Hinton broth, 25 µL of bacterial suspension, various concentrations of the test substance, and 0.9% sodium chloride solution. Wells without added bacteria served as positive controls, and wells without added substances served as negative controls. Plates were incubated for 24 h at 37 °C, and bacterial growth was assessed by the optical density of well contents using a multi-channel spectrophotometer at a wavelength of 620 nm. The optical density of the wells with test substances was compared with that of positive-control wells; a decrease in optical density compared to the positive control indicated antimicrobial activity.

### 2.7. Study of Changes in Cell Wall Uptake of Crystal Violet Dye 

Changes in absorption of the bacterial cell wall by crystal violet (CV) were examined using a modified method by Halder S. [53]. Overnight bacterial cultures were grown to exponential phase in Mueller–Hinton broth then centrifuged for 10 min at 5000 rpm. The supernatant was discarded, and the pellet was washed twice with 0.01 M phosphate buffer solution and centrifuged again under the same conditions. The pellet was resuspended in 0.01 M phosphate buffer solution to a bacterial concentration of 2 × 10^8^ CFU/mL, mixed with the test substances at a 1:1 ratio, and CV dye was added to a final concentration of 0.001%. Incubation was carried out for 30 min at 37 °C. The mixture was then centrifuged at 12,000 rpm for 2 min, the supernatant was removed, and the optical density at 630 nm was determined. The percentage of CV content in the supernatant was calculated using Formula (1):% CV content = (E_control – E_sample)/E_control × 100,(1)
where E_control is the optical density of the control sample (with test substances and CV dye), and E_ sample is the optical density of the sample with bacteria, test substances, and CV dye.

The lower the percentage of CV content in the supernatant, the better the dye was absorbed by the bacterial cell wall.

Bacterial suspensions (150 µL) were added to the wells of the plates. Negative controls consisted of wells with 150 µL of pure Mueller–Hinton broth. The plates were incubated at 37 °C for 24 h. The contents were removed from the wells, which were washed four times with distilled water. Test substances were added to the studied concentrations and Mueller–Hinton broth and incubated at 37 °C for 1 h. The wells were washed four times with distilled water, treated with 170 µL of 0.25% alcohol solution of crystal violet, and incubated at room temperature for 5 min. The plates were washed again four times with distilled water and air-dried at room temperature for 10 min. The wells were treated with 200 µL of DMSO as an extracting agent and incubated at room temperature for 10 min for complete dye extraction. The plates were then placed in a multi-channel spectrophotometer, where the optical density of the well contents was determined at a wavelength of 620 nm. Optical density values of wells with test substances were compared with those of positive-control wells, i.e., wells without test substances. A decrease in optical density compared to the positive control indicated that the test substances disrupted biofilms.

### 2.8. Assessment of Bacterial Viability

The viability of bacteria was assessed according to the protocol provided by the L-7012 LIVE/DEAD BacLight Bacterial Viability Kits (Molecular Probes, Thermo Fisher Scientific, Waltham, MA, USA). Equal volumes of component A (SYTO 9 dye) and component B (propidium iodide (PI) dye) were combined and thoroughly mixed, and 5 µL of the dye mixture was applied to a coverslip where biofilms were grown for 24 h. Incubation was carried out in the dark at room temperature for 15 min. Bacteria were observed using a confocal laser scanning microscope from Carl Zeiss, model LSM 780 NLO (Carl Zeiss Group, Oberkochen Germany), located at TRCKP TGU, at a wavelength of 488 nm. Living bacteria fluoresced green due to staining only with SYTO 9, while dead bacteria fluoresced red due to staining with propidium iodide.

### 2.9. Statistical Processing of Results

Statistical analysis was performed using “Statistica 13.0” software. The Mann–Whitney U rank test for independent samples was used to compare optical density values of control and experimental wells due to the analysis of small samples. Fisher’s exact test was used to compare the results of the inoculation cultures on Mueller–Hinton agar. Differences were considered statistically significant when they reached the 5% significance level.

Quantitative data were presented as median and interquartile range: “median (Q1; Q3)”.

## 3. Results

All synthesized phenol–humic derivatives were characterized by ^13^C NMR spectroscopy (Figure 1a,b; Table 1). NMR spectroscopy is the most powerful method for the structural analysis of organic compounds [51,54,55]. The use of NMR spectroscopy on ^13^C nuclei makes it possible to examine changes in the structure of the carbon skeleton of modified drugs compared to the original one. Figure 1a,b show the ^13^C NMR spectra of the original HSs and their phenol–humic derivatives. 

Figure 1 demonstrates that the spectra of HSs and their derivatives do not contain individual peaks and represent a superposition of a large number of signals. A qualitative comparison of the obtained spectra shows the absence of significant structural differences in the original and hydroxylated HSs. All samples (Figure 1) are characterized by a similar set and content of structural fragments. The strong overlap of the corresponding groups of signals does not allow structural differences within aromatic and aliphatic structural fragments to be spectrally distinguished.

Therefore, the obtained derivatives do not show systematic differences in their composition depending on the synthetic technique. The composition is mostly determined by the structure of the incorporated phenol into the initial humate matrix. 

Using the phenol–humic derivatives, antimicrobial active pharmaceutical substances based on silver nanoparticles were synthesized (HS-AgNPs). The list of synthesized HS-AgNPs is given in Table 2.

Direct size measurements were applied to understand the sizes of the formed AgNPs and their distributions in the solutions containing high concentrations of HSs and phenol–humic derivatives. 

The twelve Ag-HS samples prepared with the use of the higher-molecular-weight, phenolic-rich coal humate (CHP) and ten phenol–humic derivatives at the final synthesis stage yielded AgNPs. TEM images of all synthesized AgNPs in the final stable stage are shown in Figure 2. Among them, some silver nanoparticles synthesized with phenol–humic derivatives—(c) CHP-pHQ-FE-AgNPs, (d) CHP-NQ-FE-AgNPs, and (k) CHP-NQ-FF-AgNPs—and coal humate—(f) CHP-AgNPs-MW and (g) CHP-AgNPs—have a diameter of about 10 nm and a higher degree of crystallinity, whereas AgNPs synthesized with phenol–humic derivatives—(b) CHP-oHQ-FE-AgNPs, (e) CHP-HONQ-FE-AgNPs, (i) CHP-oHQ-FF-AgNPs, and (l) CHP-HONQ-FF-AgNPs—were much larger in size, ranging between 20 and 60 nm, with a predominant size of 50 nm. 

To study the antimicrobial activity of bionanomaterials (HS-AgNPs), a panel of microorganisms that are relevant in clinical practice and capable of forming biofilms was selected. The panel consisted of standard strains of *E. coli*, ATCC 25922; *S. aureus*, ATCC 25923; *MRSA*, ATCC 33592; *A. baumannii*; *P. aeruginosa*, ATCC 9027; *K. pneumoniae*, ATCC 700603; and clinical isolates of these types of microorganisms capable of biofilm formation. 

The most important samples were chosen based on the screening results (Table 3): sample No. 3 (CHP-pHQ-FE-AgNPs) against Gram-negative microorganisms (*E. coli*, *A. baumannii*, *K. pneumoniae*, and *P. aeruginosa*) and sample No. 12 (CHP-AgNPs-MW) against Gram-positive bacteria (*S. aureus* and *MRSA*). In addition, sample No. 3, which has a wide spectrum of activity on both Gram-positive and Gram-negative microorganisms, was chosen to study the effect of HS-AgNPs on the bacterial cell wall.

Sample No. 2 (CHP-oHQ-FE-AgNPs) was used at a concentration of 800 mg/L (minimum inhibitory concentration—MIC) for studying the effect of HS-AgNPs on the cell walls (*MRSA*, *K. pneumoniae*, and *P. aeruginosa*) and also used screening data (Table 3). The penetration of the crystal violet dye into the bacterial cell wall was assessed, indicating its damage—the more pronounced the damage, the more crystal violet settles in the cell wall and the less remains in the supernatant. Studying HS-AgNPs damaged the cell walls of all three bacteria studied; the greatest effect was observed for *MRSA*—the percentage of dye content in the supernatant reached 51.6 (48.4; 56.3); for *P. aeruginosa,* it was 56.6 (55.0; 57.4), and for *K. pneumoniae,* 63.7 (56.2; 65.0).

To study the effect of HS-AgNPs on the formation of biofilms by opportunistic microorganisms, the test substance was added before the start of microorganism cultivation, and the inhibition of biofilm formation was assessed by reducing the optical density of the supernatant. Biofilm formation by *E. coli*, *A. baumannii*, *K. pneumoniae*, and *P. aeruginosa* was affected by sample No. 3, and by sample No. 12 on *S. aureus* and *MRSA.*

This study aimed to investigate the impact of standard strains of opportunistic microorganisms *E. coli* and *A. baumannii* (Table 4) on biofilm formation. The results showed a significant decrease in optical density at final concentrations of the studied substances of 150 and 200 mg/L (*p* = 0.049 and *p* = 0.049, respectively). Similarly, *S. aureus* showed a decrease at a final concentration of 700, 750, and 800 mg/L (*p* = 0.049), while *MRSA* showed a decrease at a final concentration of 750 mg/L (*p* = 0.049). The optical density of *A. baumannii* at the final concentration of 100 mg/L increased (*p* = 0.049). No significant effect was observed for *K. pneumoniae* and *P. aeruginosa*.

This study focused on the inhibition of biofilm formation by clinical isolates of opportunistic microorganisms, specifically *E. coli* isolated from sputum and *A. baumannii* from urine (Table 5). The results showed a significant decrease in optical density at a final concentration of 150 mg/L (*p* = 0.049 and *p* = 0.049, respectively), and *K. pneumoniae* (urine) at final concentrations of 700, 750, and 800 mg/L (*p* = 0.049). However, for *K. pneumoniae* isolated from blood, a decrease in optical density was only seen at a final concentration of 750 mg/L (*p* = 0.049).

Sample No. 3 reduced the ability to form biofilms of standard strains of *E. coli*, *A. baumannii*, and *P. aeruginosa* and clinical isolates of *E. coli* isolated from patient sputum, *A. baumannii* from urine, and *K. pneumoniae* from urine and blood. Sample No. 12 inhibited the biofilm formation of standard strains of *S. aureus* and *MRSA*.

To study the effects of HS-AgNPs on already-formed biofilms of opportunistic microorganisms, the test substances were added after the microorganisms had been incubated for 1 h. The inhibition of biofilm formation was then assessed by measuring the decrease in the optical density of the supernatant. Sample No. 3 was tested on biofilms formed by *E. coli*, *A. baumannii*, *K. pneumoniae*, and *P. aeruginosa*, while sample No. 12 was tested on biofilms of *St. aureus* and *MRSA*.

The results of HS-AgNPs’ influence on already-formed biofilms of standard strains of opportunistic microorganisms are presented in Table 6.

The optical density of the wells’ contents after 1 h of incubation with HS-AgNPs for most of the standard strains studied increased (Table 6), which may be due to the absorption of HS-AgNPs by biofilms.

Studying the effect of HS-AgNPs on already-formed biofilms of clinical isolates of opportunistic microorganisms (Table 7), the optical density decreased in *K. pneumoniae* from patient urine at final concentrations of 700, 750, and 800 mg/L (*p* = 0.049) and *K. pneumoniae* from blood at a final concentration of 700 mg/L (*p* = 0.049).

To study the viability of bacteria in biofilms, a mixture of SYTO 9 and propidium iodide dyes was applied to a cover glass on which biofilms were grown for 24 h. The cover glass was then incubated at room temperature in the dark for 15 min and observed using a Carl Zeiss confocal laser scanning microscope at a wavelength of 488 nm. Living bacteria were stained green with SYTO 9, while dead bacteria were stained red with propidium iodide.

Standard strains of opportunistic microorganisms were studied. Sample No. 3 was used to determine the effect of nanocomposites on the viability of bacteria in the biofilm of *E. coli*, *A. baumannii*, *K. pneumoniae*, and *P. aeruginosa*, while Sample No. 12 was used for the biofilm of *S. aureus* and *MRSA*. The concentration of the HS-AgNPs was 200 mg/L for *E. coli* and *A. baumannii*, and 800 mg/L for *S. aureus*, *MRSA*, *K. pneumoniae*, and *P. aeruginosa*.

Figure 3A,B show an intact *E. coli* biofilm with a predominance of living bacteria that fluoresce green. However, when *E. coli* is incubated with sample No. 3, the biofilm structure is disrupted (Figure 3C), and there is a significant decrease in luminous cells. When viewing of the remaining liquid on a cover glass where the biofilms grew, free living bacteria and a large number of dead cells are visible (Figure 3D).

The effect of sample No. 3 on already-formed *E. coli* biofilms (Figure 3E,F) showed a decrease in cellularity compared to the control (Figure 3A,B) and a predominance of dead cells. This confirms the antibacterial effect against formed *E. coli* biofilms (Table 6).

A photo of an *A. baumannii* biofilm (Figure 3G,H) shows numerous living bacteria and a greenish glow in the biofilm’s extracellular matrix (ECM) caused by SYTO 9, which binds to nucleic acids. It is known that the ECM of biofilms contains a large amount of DNA [27]. When studying the effect of sample No. 3 on biofilm formation (Figure 3I), in addition to living cells, a large number of dead bacteria are observed. Figure 3J,K show biofilm preparations that were incubated for hour with the studied substances. The structure of the biofilm is visible, but there are significantly fewer living cells and a higher number of dead bacteria, confirming the antibacterial effect of sample No. 3 on *A. baumannii* biofilms.

Studying the effect of sample No. 12 on *S. aureus* biofilm formation, only dead bacteria were recorded (Figure 3N) compared to the control (Figure 3L,M). Interesting results were obtained when analyzing the effect of sample No. 12 on formed *S. aureus* biofilms. On the cover glass, where the biofilms were formed, no bacteria were present, but in the super-cellular fluid, entire layers of cells were found, both living and dead (Figure 3O,P). This suggests that sample No. 12 may disrupt the connection of the *S. aureus* biofilm with the substrate.

Sample No. 12 also had a suppressive effect on the formation of *MRSA* biofilm with almost complete absence of bacteria (Figure 4C) compared to the control (Figure 4A,B). The effect of sample No. 12 on *MRSA*-formed biofilm is demonstrated in Figure 4D, where the cellularity did not decrease, but a large number of dead cells were observed.

A similar effect was observed when sample No. 3 was exposed to a biofilm formed by *P. aeruginosa*, resulting in suppression of biofilm formation (Figure 4G) compared to the control (Figure 4E,F). The effect of sample No. 3 on formed *P. aeruginosa* biofilms is demonstrated in Figure 4H,I, where the biofilm structure was preserved, but the bacteria died.

The impact of sample No. 3 on *K. pneumoniae* was only evaluated with reference to the data in Table 4 and Table 6. Sample No. 3 showed antibacterial properties, as evidenced by a decrease in cellularity compared to the control (Figure 5A,B), and a high number of dead cells were observed (Figure 5C,D).

Thus, it can be concluded that the viability of the studied bacteria in biofilms is reduced when they are exposed to bionanomaterials containing ultradispersed silver nanoparticles in a humic substance matrix.

## 4. Discussion

The creation of alternative antimicrobial drugs based on silver nanoparticles is of serious scientific interest since it allows for a reduction in the concentration of the active substance many times over while maintaining its bactericidal properties [56,57]. Silver nanoparticles have the characteristics necessary to combat microorganisms: they are stable, in a colloidal state, and have good reactivity. Their nanosize ensures greater contact with the pathogenic microorganism due to the surface area of the particles and better bioavailability, causing the death of bacteria even in low concentrations [43]. Numerous studies have demonstrated the pronounced antibacterial activity of silver nanoparticles against pathogenic clinical Gram-positive and Gram-negative bacterial strains [58,59,60,61]. The mechanism of action of silver nanoparticles is complex, implemented in several ways with a mandatory effect on the cell wall of the microbe. In addition, it has been proven that silver nanoparticles can penetrate the cytoplasmic membranes of microbial cells [62]. However, most studies are devoted to the direct bactericidal effect of silver nanoparticles on various microorganisms [63,64,65]. However, no less significant is the ability of nanocomposites to influence the biofilm formation of bacterial pathogens. Thus, K. Niska et al. [66,67] discovered the suppression of biofilm formation by pathogenic bacteria in the oral cavity under the action of silver nanoparticles. The authors note that the stabilization of nanoparticles with lipoic acid reduces the cytotoxicity of the drug and increases the antibacterial effect [66,67] and proved the antimicrobial activity of silver nanoparticles against mature biofilms of *S. mutans*. J.M. Corrêa et al. [68] even suggested including silver nanoparticles in composite resins and adhesive systems to prevent biofilm formation by pathogens of dental diseases. Humic substances as source matrices can enhance and complement the effectiveness of silver nanoparticles. Microorganisms can use humic substances as a source of nutrients [42]. The bioavailability of metals in soils is determined by their complexation with humic acids, which makes these metals more bioavailable to microorganisms. Based on our findings and existing research on humic substances, it can be inferred that the studied bionanomaterials are sorbed onto the CPM and absorbed by bacteria, enabling silver nanoparticles to exert their antimicrobial effects.

Bionanomaterials based on silver nanoparticles ultradispersed in a matrix of humic substances are promising in terms of their antimicrobial effect. To influence microorganisms, the presence of silver is primarily important due to its antibacterial properties.

It is also worth noting that free silver nanoparticles can have a cytotoxic effect on the body’s cells [69], while humic substance matrices can eliminate this undesirable effect [70]. An in vitro cytotoxicity study of HS-AgNPs samples was previously conducted and found that stabilization of AgNPs using HS matrices reduces the cytotoxicity of AgNPs [30].

The combination of silver nanoparticles with humic substances increases the chances of nanocomposites penetrating both the bacteria themselves and their biofilms because there is evidence of the humic substances’ ability to penetrate cells [39] due to their amphiphilicity and surface-active properties [70].

In addition, the void that is formed as a result of the bacteria’s death inside the biofilm can contribute to its subsequent detachment [71]. The destruction of biofilms is enhanced by the rejection of their fragments. Microorganisms lose their protective shell in the form of a biofilm and become more accessible to the same silver nanoparticles or other antibiotics. This may become the basis for the use of combination antimicrobial therapy, using the possibility of reducing drug doses due to joint action.

In addition, humic substances can form interionic bonds with high-molecular-weight structural polymers of bacteria [72]. One of the most important components of the extracellular matrix of the biofilm is exopolysaccharides. They play a key role in ECM formation and microorganism community resistance to antimicrobial agents. It is possible that bio-nanomaterials based on silver nanoparticles, ultradispersed in the matrix of humic substances, bind specifically to exopolysaccharides, thereby destroying the architecture of the biofilm and its protective properties and gaining the ability to penetrate the bacteria themselves.

One of the features of biofilm as a form of bacteria existence is its tight attachment to the substrate. The impact of testing bionanomaterials is that it disrupts the strength of these bonds, the matrix ceases to perform its functions, the structure of the biofilm is disrupted, and the bacteria die. A clear confirmation of this is the results of assessing the viability of bacteria obtained using confocal laser scanning microscopy. Studying the effect of HS-AgNPs on the biofilm’s formation, samples were introduced into the wells of the plate simultaneously with a suspension of bacteria, so they could directly influence the processes of formation of biofilm structures. The cellularity of these preparations is much less than in intact biofilms, or approximately the same, but with a larger number of dead bacteria. This may indicate the ability of the HS-AgNPs to damage the cell walls of bacteria.

The results obtained with other methods of assessing the effect of HS-AgNPs on biofilms coincide with the results of assessing viability. The optical density of the well contents decreased when assessing the effect of substances on the formation of biofilms of all clinical and almost all studied standard strains, which indicates a weaker formation of biofilms by them. A study of the effect of HS-AgNPs on the bacterial cell wall showed the absorption of the CV dye by both Gram-positive MRSA and Gram-negative K. pneumoniae and P. aeruginosa, which indicates the ability of HS-AgNPs to damage the bacterial cell wall.

It is worth noting other properties of bionanomaterials based on silver nanoparticles ultradispersed in a matrix of humic substances. They have proven antioxidant activity [30,73] and immunomodulatory properties [74]. This allows us to consider the use of bionanomaterials based on silver nanoparticles ultradispersed in a matrix of humic substances for the treatment of various infectious processes.

## 5. Conclusions

Thus, bionanomaterials based on silver nanoparticles ultradispersed in the matrix of humic substances demonstrated antibacterial properties against all selected opportunistic microorganisms capable of forming biofilms. The panel included both Gram-positive and Gram-negative microorganisms, including standard and clinical strains such as *Escherichia coli*, ATCC 25922; *Staphylococcus aureus*, ATCC 25923; *MRSA*, ATCC 33592; *Klebsiella pneumoniae*, ATCC 700603; and *Pseudomonas aeruginosa*, ATCC 9027. An inhibitory effect was established both on the process of biofilm formation of these bacteria and on already-formed biofilms. The experimental sample CHP-pHQ-FE-Ag (No. 3) showed the highest antibacterial activity against Gram-negative strains—*E. coli*, *A. baumannii*, *K. pneumoniae*, and *P. aeruginosa*. The experimental sample CHP-AgNPs-MW (No. 14) showed the highest antibacterial activity against Gram-positive strains, specifically *St. aureus*, *MRSA*. 

The particular advantage of the proposed antibacterial nanomaterials is a straightforward approach to scaling up of their production. We used a well-established and industrially proven technique, microwave heating, which enables efficient and speedy large-scale production of these materials. Another advantage is a lack of waste as both silver-containing precursor and humate are used in full for production of the nanomaterials. No other auxiliary materials or compounds are used; just alkali and water are added to the precursors to aid production of the nanomaterial. These specific features of HS-based nanosilver compositions are very attractive from the point of view of the possible environmental impact of their production. 

The limitation of the proposed antibacterial composition is the very high concentration of silver, which is needed for maintaining its antibacterial properties. However, this limitation is inherent within all silver, and nanosilver-based antibacterial compositions require higher concentrations due to the much lower bactericidal activity of silver compared to synthetic antibiotics. To overcome this important limitation, which is intimately connected to another adverse environmental impact, the accumulation of nanosilver in the ecosystem, the promising direction is in designing silver-based nanomaterials with very high antibacterial activity by combining antibiotics and silver nanoparticles. There are a lot of recent reports on synergistic activities of antibiotics and nanosilver [75,76]. We have also demonstrated the promising combination of lincomycin and HS-based silver nanomaterials for restoring sensitivity of the MRSA strain to lincomycin [31]. We believe that this direction might be the most productive for combating biofilm-forming pathogens with the use of nanomaterials. The synergistic effects of antibiotics and nanosilver drastically reduce the dosage of silver in these compositions and resolve one of the biggest limitations of silver-based nanomaterials along with maintaining and increasing their effectiveness.

## Figures and Tables

**Figure 1 nanomaterials-14-01453-f001:**
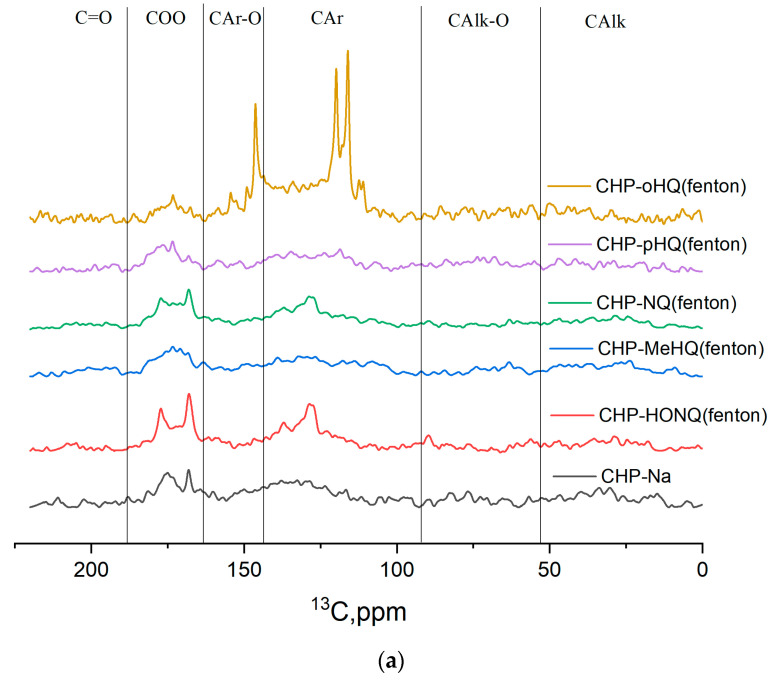
^13^C NMR spectra of the original preparation of HSs and their phenol–humic derivatives: (**a**) synthesized using Fenton’s reagent; (**b**) synthesized using phenol–formaldehyde condensation.

**Figure 2 nanomaterials-14-01453-f002:**
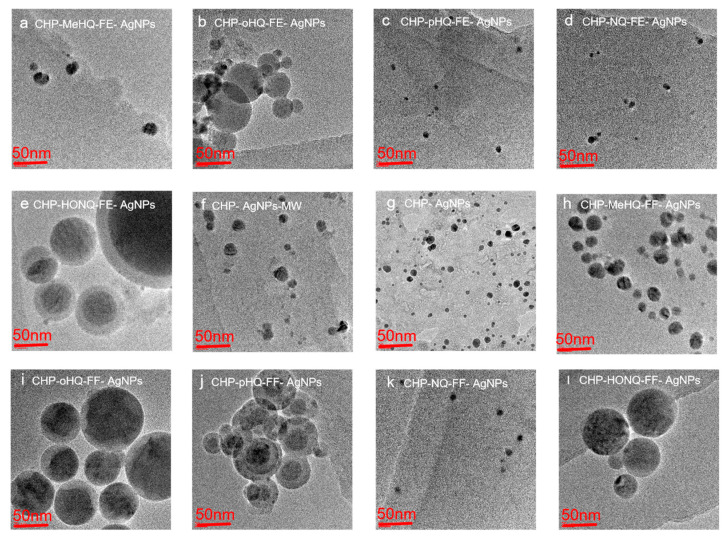
The particle size and distribution of AgNPs synthesized in the presence of coal humate (CHP) and ten phenol–humic derivatives: synthesized using Fenton’s reagent (**a**) CHP-MeHQ-FE-AgNPs, (**b**) CHP-oHQ-FE-AgNPs, (**c**) CHP-pHQ-FE-AgNPs, (**d**) CHP-NQ-FE-AgNPs, and (**e**) CHP-HONQ-FE-AgNPs; synthesized using phenol–formaldehyde condensation (**h**) CHP-MeHQ-FF-AgNPs, (**i**) CHP-oHQ-FF-AgNPs, (**j**) CHP-pHQ-FF-AgNPs, (**k**) CHP-NQ-FF-AgNPs, and (**l**) CHP-HONQ-FF-AgNPs; under conditions of MW-assisted heating (**f**) CHP-AgNPs-MW; and under conditions of conventional heating (**g**) CHP-AgNPs.

**Figure 3 nanomaterials-14-01453-f003:**
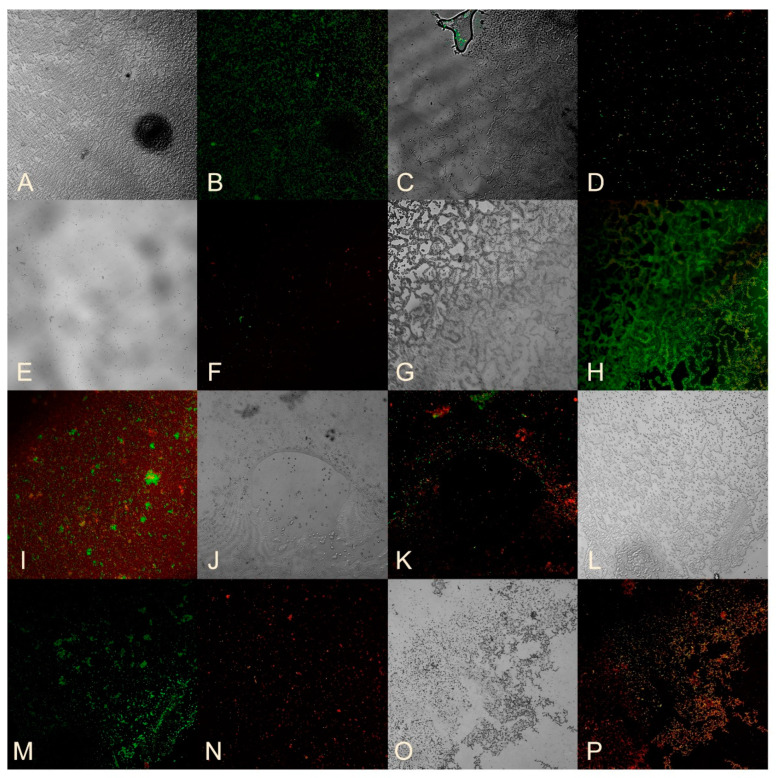
(**A**) Standard strain of *E. coli* biofilm. (**B**) Visualization of *E. coli* bacteria using 510–540 nm and 620–650 nm filters. (**C**) *E. coli* biofilm formed in a medium with the addition of HS-AgNPs. (**D**) *E. coli* in liquid flowing over a coverslip. Preparation of *E. coli* biofilm formed in a medium with the addition of HS-AgNPs. (**E**) *E. coli* biofilm incubated for 1 h with the HS-AgNPs. (**F**) Visualization of *E. coli* bacteria using 510–540 nm and 620–650 nm filters. (**G**) Biofilm of standard strain of *A. baumannii*. (**H**) Visualization of *A. baumannii* bacteria using 510–540 nm and 620–650 nm filters. (**I**) *A. baumannii* biofilm formed in a medium with the addition of HS-AgNPs. (**J**) *A. baumannii* biofilm incubated for 1 h with the HS-AgNPs. (**K**) Visualization of *A. baumannii* bacteria using 510–540 nm and 620–650 nm filters. (**L**) Biofilm of standard strain of *S. aureus*. (**M**) Visualization of *S. aureus* bacteria using 510–540 nm and 620–650 nm filters. (**N**) *S. aureus* biofilm formed in a medium with the addition of HS-AgNPs. (**O**) *S. aureus* biofilm incubated for 1 h with the HS-AgNPs on a microscopy slide. (**P**) Visualization of *S. aureus* bacteria using 510–540 nm and 620–650 nm filters. ×300.

**Figure 4 nanomaterials-14-01453-f004:**
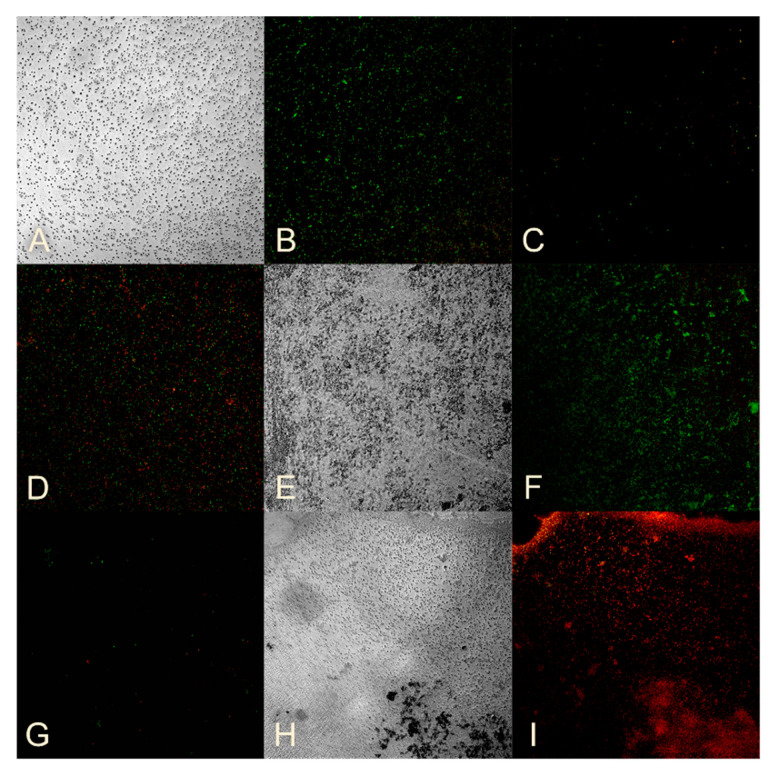
(**A**) Standard strain of *MRSA* biofilm. (**B**) Visualization of *MRSA* bacteria using 510–540 nm and 620–650 nm filters. (**C**) *MRSA* biofilm formed in a medium with the addition of HS-AgNPs. (**D**) *MRSA* biofilm incubated for 1 h with the HS-AgNPs. (**E**) Standard strain of *P. aeruginosa* biofilm. (**F**) Visualization of *P. aeruginosa* bacteria using 510–540 nm and 620–650 nm filters. (**G**) *P. aeruginosa* biofilm formed in a medium with the addition of HS-AgNPs. (**H**) *P. aeruginosa* biofilm incubated for 1 h with the HS-AgNPs. (**I**) Visualization of *P. aeruginosa* bacteria using 510–540 nm and 620–650 nm filters. ×300.

**Figure 5 nanomaterials-14-01453-f005:**
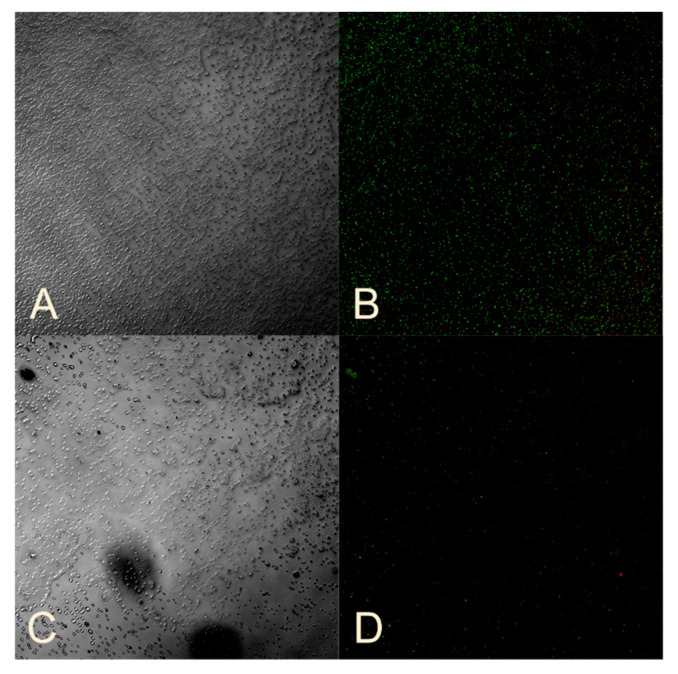
(**A**) Standard strain of *K. pneumoniae* biofilm. (**B**) Visualization of *K. pneumoniae* bacteria using 510–540 nm and 620–650 nm filters. (**C**) *K. pneumoniae* biofilm incubated for 1 h with the HS-AgNPs. (**D**) Visualization of *K. pneumoniae* bacteria using 510–540 nm and 620–650 nm filters. ×300.

**Table 1 nanomaterials-14-01453-t001:** Structural-group composition of the phenol–humic derivatives prepared from the humic substances with the use of two different synthetic techniques.

Samples	CH	CHO	C_ar_	C_ar_O	COO	C=O
	0–47	47–110	110–145	145–165	165–185	185–220
CHP-Na	17.5	18.7	34.4	9.8	20.6	4.7
CHP-HQ-FF	14.5	32.6	39.0	11.3	13.7	1.2
CHP-HQ-FE	16.7	31.6	22.4	10.1	27.6	4.7
CHP-HONQ-FF	11.5	34.9	34.5	11.8	17.0	4.2
CHP-HONQ-FE	13.3	24.6	33.9	9.0	24.0	4.3
CHP-MeHQ-FF	18.6	36.8	37.5	9.6	8.9	2.9
CHP-MeHQ-FE	19.3	31.3	25.7	9.8	18.2	8.0
CHP-PK-FF	15.5	24.3	40.0	9.9	15.3	4.2
CHP-PK-FE	9.0	30.1	49.5	13.4	6.7	4.0
CHP-NQ-FF	17.0	19.7	43.0	8.4	16.4	2.0
CHP-NQ-FE	14.6	44.4	28.7	7.6	19.0	4.7

**Table 2 nanomaterials-14-01453-t002:** List of samples of bionanomaterials (HS-AgNPs) based on silver nanoparticles ultradispersed in a matrix of humic substances.

No.	Code	Derivative for HS-AgNPs Synthesis	Reaction for HS-AgNPs Synthesis	Ag Content, mg/100 mL
1	CHP-AgNPs	-	Template synthesis	24.27
2	CHP-oHQ-FE-AgNPs	*o-*hydroquinone	Fenton	24.27
3	CHP-pHQ-FE-AgNPs	*p-*hydroquinone	Fenton	24.27
4	CHP-MeHQ-FE-AgNPs	2-methyl-1,4-hydroquinone	Fenton	24.27
5	CHP-NQ-FE-AgNPs	1,4-naphthoquinone	Fenton	24.27
6	CHP-HONQ-FE-AgNPs	2-hydroxy-1,4-naphthoquinone	Fenton	24.27
7	CHP-oHQ-FF-AgNPs	*o-*hydroquinone	Phenol–formaldehyde condensation	24.27
8	CHP-pHQ-FF-AgNPs	*p-*hydroquinone	Phenol–formaldehyde condensation	24.27
9	CHP-MeHQ-FF-AgNPs	2-methyl-1,4-hydroquinone	Phenol–formaldehyde condensation	24.27
10	CHP-NQ-FF-AgNPs	1,4-naphthoquinone	Phenol–formaldehyde condensation	24.27
11	CHP-HONQ-FF-AgNPs	2-hydroxy-1,4-naphthoquinone	Phenol–formaldehyde condensation	24.27
12	CHP-AgNPs-MW	-	Microwave synthesis	24.27

**Table 3 nanomaterials-14-01453-t003:** The active pharmaceutical substances (HS-AgNPs) that suppress the growth of opportunistic microorganisms.

Microorganisms	Concentration of HS-AgNPs That Suppressed the Growth of Microorganisms		
	200 mg/L	500 mg/L	800 mg/L
*E. coli*	No. 2, No. 3	No. 2, No. 3	No. 2, No. 3, No. 4, No. 6, No. 7
*MRSA*	–	–	No. 12
*K. pneumoniae*	–	–	No. 3
*P. aeruginosa*	No. 3	–	No. 3, No. 5, No. 6, No. 7
*A. baumannii*	No. 2, No. 3, No. 4, No. 6	No. 1, No. 2, No. 3, No. 4, No. 6, No. 78, No. 12	No. 1, No. 2, No. 3, No. 4, No. 6, No. 7, No. 8, No. 10, No. 12

**Table 4 nanomaterials-14-01453-t004:** The influence of the active pharmaceutical substances (HS-AgNPs) on the formation of biofilms by standard strains of opportunistic microorganisms.

Bacteria	Sample	Concentration of HS-AgNPs	Optical Density	p
*E. coli* (N = 3)	No. 3	100	0.164 (0.164; 0.228)	0.513
		150	0.141 (0.125; 0.144)	0.049
		200	0.170 (0.133; 0.181)	0.049
		Control	0.216 (0.195; 0.226)	
*A. baumannii* (N = 3)	No. 3	100	0.348 (0.345; 0.348)	0.049
		150	0.317 (0.308; 0.326)	0.049
		200	0.316 (0.306; 0.318)	0.049
		Control	0.330 (0.329; 0.330)	
*K. pneumoniae* (N = 3)	No. 3	700	0.314 (0.307; 0.316)	0.513
		750	0.349 (0.337; 0.365)	0.513
		800	0.293 (0.273; 0.300)	0.513
		Control	0.321 (0.237; 0.392)	
*P. aeruginosa* (N = 3)	No. 3	700	0.189 (0.184; 0.190)	0.513
		750	0.186 (0.179; 0.193)	0.513
		800	0.178 (0.170; 0.194)	0.513
		Control	0.248 (0.169; 0.262)	
*S. aureus* (N = 3)	No. 12	700	0.206 (0.205; 0.236)	0.049
		750	0.195 (0.192; 0.205)	0.049
		800	0,198 (0.197; 0.199)	0.049
		Control	0.273 (0.261; 0.284)	
*MRSA* (N = 3)	No. 12	700	0.270 (0.261; 0.276)	0.513
		750	0.245 (0.241; 0.251)	0.049
		800	0.254 (0.242; 0.257)	0.127
		Control	0.341 (0.255; 0.372)	

Note: N—number of replicates; p—significance level.

**Table 5 nanomaterials-14-01453-t005:** The influence of the active pharmaceutical substances (HS-AgNPs) on the formation of biofilms by clinical isolates of opportunistic microorganisms.

Bacteria	Material	Sample	Final Concentration of HS-AgNPs, mg/L	Optical Density	p
*E. coli* (N = 3)	Sputum	No. 3	100	0.129 (0.119; 0.140)	0.275
			150	0.111 (0.104; 0.120)	0.049
			200	0.127 (0.127; 0.132)	0.127
			Control	0.139 (0.130; 0.215)	
*A. baumannii* (N = 3)	Urine	No. 3	100	0.274 (0.255; 0.294)	0.827
			150	0.224 (0.213; 0.227)	0.049
			200	0.243 (0.238; 0.271)	0.275
			Control	0.259 (0.250; 0.347)	
*K. pneumoniae* (N = 3)	Urine	No. 3	700	0.166 (0.158; 0.195)	0.049
			750	0.206 (0.204; 0.229)	0.049
			800	0.210 (0.172; 0.215)	0.049
			Control	0.270 (0.244; 0.300)	
*K. pneumoniae* (N = 3)	Blood	No. 3	700	0.207 (0.207; 0.231)	0.127
			750	0.207 (0.200; 0.222)	0.049
			800	0.234 (0.223; 0.241)	0.275
			Control	0.260 (0.227; 0.328)	

Note: N—number of replicates; p—significance level.

**Table 6 nanomaterials-14-01453-t006:** The influence of the active pharmaceutical substances (HS-AgNPs) on the formation of biofilms of standard strains of opportunistic microorganisms.

Bacteria	Sample	Final Concentration of HS-AgNPs, mg/L	Optical Density	p
*E. coli* (N = 3)	No. 3	100	0.232 (0.204; 0.245)	0.049
		150	0.231 (0.224; 0.238)	0.049
		200	0.232 (0.231; 0.238)	0.049
		Control	0.181 (0.169; 0.193)	
*A. baumannii* (N = 3)	No. 3	100	0.335 (0.316; 0.346)	0.049
		150	0.326 (0.315; 0.328)	0.049
		200	0.329 (0.327; 0.336)	0.049
		Control	0.232 (0.168; 0.308)	
*S. aureus* (N = 3)	No. 12	700	0.290 (0.284; 0.305)	0.049
		750	0.296 (0.285; 0.300)	0.049
		800	0.292 (0.286; 0.304)	0.049
		Control	0.227 (0.224; 0.249)	
*MRSA* (N = 3)	No. 12	700	0.367 (0.363; 0.373)	0.049
		750	0.346 (0.324; 0.350)	0.049
		800	0.370 (0.345; 0.386)	0.049
		Control	0.266 (0.216; 0.308)	
*K. pneumoniae* (N = 3)	No. 3	700	0.346 (0.256; 0.351)	0.513
		750	0.365 (0.360; 0.373)	0.049
		800	0.337 (0.335; 0.381)	0.049
		Control	0.322 (0.305; 0.333)	
*P. aeruginosa* (N = 3)	No. 3	700	0.309 (0.283; 0.310)	0.049
		750	0.323 (0.292; 0.330)	0.049
		800	0.266 (0.237; 0.276)	0.513
		Control	0.252 (0.244; 0.260)	

Note: N—number of replicates; p—significance level.

**Table 7 nanomaterials-14-01453-t007:** The influence of the active pharmaceutical substances (HS-AgNPs) on formed biofilms of clinical isolates of opportunistic microorganisms.

Microorganism	Material	Sample	Final Concentration of HS-AgNPs, mg/L	Optical Density	p
*E. coli* (N = 3)	Sputum	No. 3	100	0.197 (0.181; 0.216)	0.513
			150	0.185 (0.179; 0.190)	0.513
			200	0.195 (0.157; 0.196)	0.663
			Control	0.157 (0.142; 0.235)	
*A. baumannii* (N = 3)	Urine	No. 3	100	0.309 (0.304; 0.314)	0.513
			150	0.302 (0.293; 0.319)	0.513
			200	0.293 (0.286; 0.297)	0.512
			Control	0.329 (0.274; 0.375)	
*K. pneumoniae* (N = 3)	Urine	No. 3	700	0.265 (0.236; 0.328)	0.049
			750	0.266 (0.247; 0.273)	0.049
			800	0.260 (0.255; 0.281)	0.049
			Control	0.489 (0.456; 0.553)	
*K. pneumoniae* (N = 3)	Blood	No. 3	700	0.249 (0.244; 0.251)	0.049
			750	0.271 (0.260; 0.288)	0.513
			800	0.262 (0.251; 0.279)	0.275
			Control	0.278 (0.269; 0.390)	

Note: N—number of replicates; p—significance level.

## Data Availability

Data are contained within the article.

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
