# Peer review of "The Influence of Silver-Containing Bionanomaterials Based on Humic Ligands on Biofilm Formation in Opportunistic Pathogens"

_nanomaterials, 2024, doi:10.3390/nano14171453_

Round 1
Reviewer 1 Report (Previous Reviewer 1)
Comments and Suggestions for Authors
I have additional minor point: in abstract and subtitles Figs. 3,4,5 all bacterial Latin names should be italics. In abstract line 27 should be S. aureus, line 28 should be K. pneumonia, S. aureus
Author Response
Please, find the attached file

Reviewer 2 Report (Previous Reviewer 2)
Comments and Suggestions for Authors
Upon reviewing what appears to be a revised version of the manuscript, it appears that there have been significant improvements in terms of highlighted texts and grammar corrections. However, as I did not have the opportunity to assess the initial version, I am unable to gauge the progress made in the review process. Nevertheless, I would like to offer some suggestions for enhancement:
Introduction:
The study discusses the antibacterial and biofilm-inhibiting properties of silver nanoparticles, but it would benefit from a more detailed exploration of the mechanistic interactions of these nanoparticles at the molecular level with bacterial cells and biofilms.
Results:
While the focus on specific opportunistic pathogens is relevant, further studies may be necessary to ascertain the generalizability of the findings to a wider range of bacteria.
Discussion:
The potential environmental impact of silver nanoparticles, particularly in terms of disposal and accumulation in ecosystems, warrants consideration and should be included in the discussion.
The long-term stability and efficacy of the bionanomaterials, crucial for practical applications, should be addressed in the study.
Furthermore, regulatory and ethical considerations pertaining to the use of such nanomaterials in medical applications are lacking in the discussion and should be incorporated.
Author Response
Please, find the attached file

Reviewer 3 Report (Previous Reviewer 3)
Comments and Suggestions for Authors
The authors revised and updated their manuscripts properly, the MS is feasible to publish now.
Comments on the Quality of English LanguageNow good to publish, but better to perform a thorough revision during a proofreading
Author Response
Please, find the attached file

Reviewer 4 Report (New Reviewer)
Comments and Suggestions for Authors
Please find attached the review report with comments and suggestions for the authors

Please find attached the review report with comments and suggestions for the authors
Author Response
Please, find the attached file

Round 2
Reviewer 2 Report (Previous Reviewer 2)
Comments and Suggestions for Authors
Congratulations for the good work.
This manuscript is a resubmission of an earlier submission. The following is a list of the peer review reports and author responses from that submission.
Round 1
Reviewer 1 Report
Comments and Suggestions for Authors
For authors:
This article contain sufficient information relevant to Nanomaterials Journal (results and theoretical). Taking into account importance and extent of undertaken subject the authors should consider the following points:
· The novelty of synthesis of HS and silver nanoparticles should be explained in introduction section.
· Is CV dyes means crystal violet? (line 252)
· · Even though the authors do a very good job in describing the structural analysis of organic compounds, an characterization of nanoparticles (silver nanoparticles size, BET, SEM images) is completely missing.
· · In materials and methods the authors suggests that the percentage of CV content in the supernatant was calculated but I did not find the results in presented tables.
· · What means N=3 and p in Table 4, 5,6 and 7?
· · I do not understand what means the value in brackets e.g. 0.321 (0.237; 0.392). In my opinion it should be explained in results tables.
· · I do not understand what means the value in brackets. For example line 342” percentage of dye content in the supernatant reached 51.6 (48.4; 56.3). The mean average for the values in brackets is 52.35?
· · Figure 3-20 The authors should indicate what kind of strain was used (standard strain or clinical isolate)?
· · The discussion section need improvement. In this section, the Authors did not discuss the relationship (agreement, disagreement, extension) of own results to other published findings (in particular this concerns the reduction of the formation of bacterial biofilms by others silver nanoparticles).
Minor corrections are also needed:
Line 212, 214, 252, should be 1.5x10-8 CFU/mL (minus in the power)
Line 110, 335, 337, 571-573 should be Gram-positive and Gram-negative
Reviewer 2 Report
Comments and Suggestions for Authors
Thank you for inviting me to review this interesting manuscript. The study focused on bionanomaterials based on silver nanoparticles ultradispersed in a matrix of humic substances (HS-AgNPs) and their effects on biofilm formation by opportunistic pathogens. This a noteworthy project however several improvements are required before it can be acceptable for publication.
Major issues:
1. The study often uses only 3 replicates (N=3) for experiments, which may not be sufficient for robust statistical analysis and could limit the reliability of the results.
2. The research doesn't compare the effectiveness of HS-AgNPs to conventional antibiotics or other known antimicrobial agents, making it difficult to assess their relative efficacy.
Minor issues:
While the study suggests possible mechanisms of action, it doesn't provide definitive evidence for how the HS-AgNPs interact with bacteria and biofilms at a molecular level. Please extend more on that.
Also, the study tests various concentrations but doesn't provide a clear rationale for optimal dosage for different pathogens.
Comments on the Quality of English Language
There are some areas where the English could be improved for clarity, which might impact the understanding of some technical details. Areas for improvement:
Misuse of articles (a/an/the) in several instances. Occasional subject-verb agreement errors,
silver containing should be silver-containing,
ones ability instead of one's ability,
Inconsistent use of tenses, especially when describing methods and results and some sentences lacking proper parallel structure.
Consider seeking the assistance of a native English speaker or a professional editing service if needed. A well-polished manuscript will greatly improve its chances of acceptance and ensure that your important findings are effectively communicated to the scientific community.
Reviewer 3 Report
Comments and Suggestions for Authors
1. Abstract: Too long, abstract shall be concise, directly indicating what is the background, shortcomings of current intervention, and the novelty of your study in the abstract. Never start your abstract like a narrative story, like "some experiments were performed", also you might divide your abstract into these sections: backgrounds, methods, findings, novelty and how you bridged the research gaps, and the conclusions. Please rewrote your abstract completely.
2. Abstract and MS: Please try to utilize passive forms when writing academic manuscripts, try to avoid "This article describes..." "we found that"...
3. Introduction, Line 41:“ antagonism of antibiotics” is not very accurate, better utilize "antibiotics resistance"
4. Introduction, line 55-58, sentence is too long and difficult to read, better make it simple, e.g “ The WHO concerned that without appropriate measurements to curb AMR, a heavy financial burden of approximately $100 trillion might be loaded on the global GDP and over 10 million patients might be thereatened by premature deaths.” Things like that.
5. Introduction is too long meanwhile the discussions is too short, actually more details shall be presented in the discussion, please consider shortening the introduction, and move some paragraphs to the discussion section.
6. Introduction, line 92-94: HS are good ligands??? Please do note that academic manuscripts never tolerate such subjective assertions. If it is good, then please write some details on why they are good here?
7. Paragraph 1-2, consider shorten, put into 1, and put some into the discussion, also combine the paragraph 3-5 as they were just like fragments.
8. Methods and materials: 2.1, 2.2, 2.4, three synthesis of materials please combine into one
9. Methods: why 2.6 is that long??? Please shorten that as the length of 2.5, if can't, divide into 2-3 section.
10. Figure 1,2 consider combine into 2.
11. Table 4-6: Those results will be more straightforward should you present them into the bar graph, please consider make several bar graphs for them
12. Figure 3-20: Please combine those figures, never include 1-2 images as a single figure, consider combining them into 1-2 figures, maximum 3.
13. Discussion: Too short with insufficient information presented, discussion should be the most intriguing section of your paper, and shall be expanded, at least twice the length of your present one, the following suggestions tells you how you could expand it:
14. Discussion: Combine paragraph 1-3 into 1, following this sequence will make the flow of story better: first paragraph 3: Microorganisms are not able to XXX, then paragraph 1: Silver XXX, finally the paragraph 2: combination of silver with NPs..... Please write more things here, cite more papers. one paper is suggested here [Guo et al. Dent Mater. 2021, Race to invade.....]
15. Discussion: Paragraph 4, 5, please combine into 1, and please don't utilize "in addition" too frequently, use other synonyms such as "moreover, furthermore, subsequently"
16. Discussion: last paragraph shall be a small conclusion of your main findings with the novelty, why you utilize that design, why you design your study like this, what shall it bring to us?
17. Discussion: the 2nd last paragraph shall be a limitation of your current study, what shall be improved, and what could be performed to improve your study if you were asked to start another project.
18. Conclusion: "all studied opportunistic microorganisms", consider rewrote that, actually you selected different types of microorganisms, could they reflect several types of different bacteria or represent all, or most or even a large part of bacteria involved in infectious diseases? like that, What I suppose is that please write some conclusive, constructive words here.
Comments on the Quality of English LanguageEnglish shall be extensively improved, and should be thoroughly reviewed by a native speaker, at least by a well-trained PhD.
Tense shall be consistent, past forms is suggested, and selection of words shall be careful when writing academic manuscripts.